# SemCLIP: Aligning vision-language encoder models to semantic spaces for stability in retrieval

## Abstract

Vision-language models (VLM) bring image and textual representations close together in a joint embedding space to tackle many tasks ranging from image captioning to retrieval. For such models to be reliably used in cloud vector stores, it is important to have a stable association between images and text such that synonymous queries bring up the same images or have a high degree of overlap. Current textual representations based on transformer models used to build the VLMs, cannot adequately capture linguistic similarities to ensure such stability. In this paper, we develop a dataset of linguists-curated similarity lists of words derived from Wordnet and train a semantics preserving textual embedding (STE). We then train an alignment transformation to map existing VLM (CLIP) embeddings to the STE embeddings to bring synonymous text and their associated images closer while preserving image-text similarities. The alignment transform is learned from textual embeddings alone thus avoiding large-scale retraining of VLMs from image-text pairs. This simple method surprisingly outperforms other methods of creating image-joint text embeddings including those by fine-tuning the encoders using the same synonym lists as evaluated on multiple benchmark datasets. The dataset of similarity lists and the semantics-preserve textual embedding itself can be employed in a variety of ways for other downstream tasks and will be made available for other researchers.

## 1 Introduction

More and more enterprises are opting for cloud vector databases for storing and managing data such as photos, video, audio, and documents (Weaviate, 2024; Databricks, 2024; Snowflake, 2024). In these, the content is stored as vectors and retrieved with vectors formed from textual queries through vision-language models (VLMs)(He et al., 2017; Radford et al., 2021; Gu et al., 2021; Zareian et al., 2021; Hinami & Satoh, 2018). For such VLM models to be reliably used in cloud vector stores, it is important to have a stable association between images and text such that synonymous queries bring up the same images or have a high degree of overlap. Currently, the VLM models derive the joint image-text embedding starting from encodings of associated text based on transformer model variants (Radford et al., 2021; Li et al., 2022). The transformer models are trained from data in a self-supervised way and are good for capturing semantic similarity of terms in use context rather than an explicit recognition of linguistic similarities through synonymous words across sentences (Chang et al., 2020; Reimers & Gurevych, 2019a). This leads the generated VLMs to also lose this sensitivity leading to instabilities in downstream uses such as text-to-image retrieval.

Figure 1 illustrates this problem, showing examples of the top 5 retrieved images from sets of similar queries prompted by "Images of X" where X is the phrase on top of each column. In Figure 1(a)-(b), synonymous terms "hamper" and "basket" retrieve different top 5 matches. This problem is also seen when more context is available as in the queries of Figure 1(c)-(d) where more terms are replaced by their synonymous phrases (overcoat→coat, frock→gown). Table 1 further illustrates the fact that the loss of sensitivity to linguistic similarity is due to the underlying textual embeddings such as BERT (Devlin et al., 2019). Table 1, Column 1 shows a group of words that are synonyms of the word "Orbital module" which have low cosine similarity using a popular text encoding based on sentence BERT (SBERT) (Reimers & Gurevych, 2019b) as well as CLIP VLM model (Radford

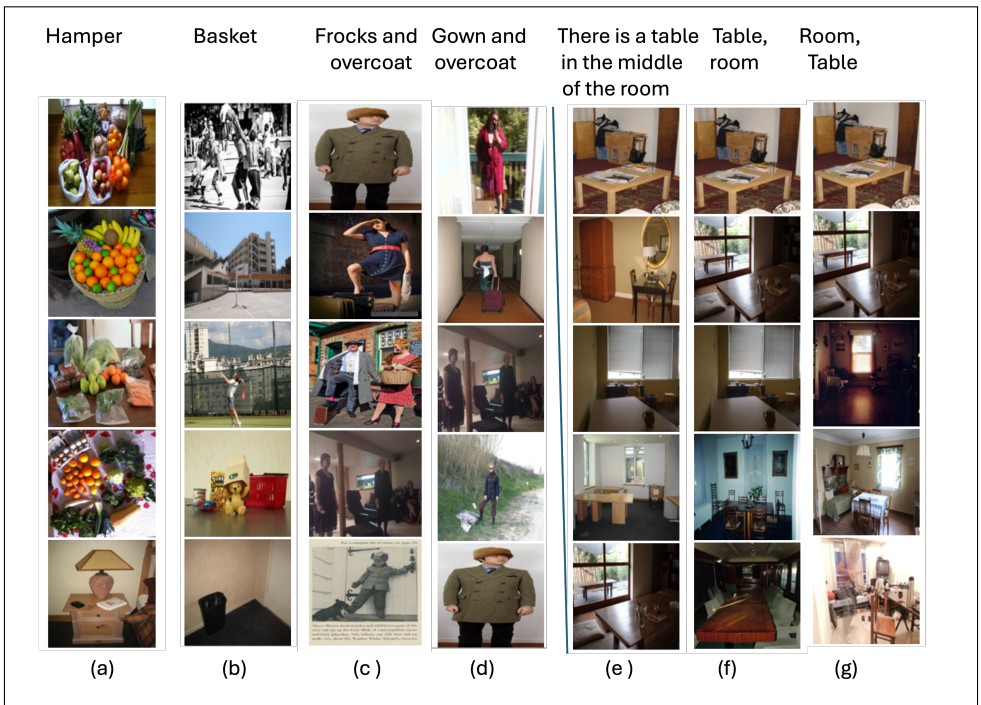

Figure 1: Illustration of retrieval instability to synonymous phrases in vision language models. (a)-(b) for isolated synonymous words. (c)-(d) for phrases. (e)-(g) shows higher overlap when full phrase is replaced by nouns only indicating approximating by nouns within query is sufficient.

et al., 2021). Column 4 of Table 1, on the other hand shows sentences that are semantically different in a subtle way in key places, but the large lexical overlap makes them highly similar in both SBERT and CLIP embeddings.

Thus there is a need for new textual embeddings that can better capture the linguistic similarity in terms *and* a method to align the current VLM models with such textual embeddings, preferably without large-scale re-training of the VLM models. In this paper we propose such an approach that achieves this alignment through 3 key novel methodological and dataset contributions:

- We develop a database of 114,000 linguists-curated similarity lists of words from a constrained traversal of Wordnet thesaurus to cover all English language nouns and use a representation to capture their sense context explicitly. All datasets produced in the paper will be contributed back to the community via open source.

- We train a semantics-preserving textual embedding (STE) using over 600,000 pairs of synonyms terms derived from these similarity lists to discover expanded synonymous relations between terms through inference using supervised contrastive learning.

- We then develop a method to align a VLM embedding such as CLIP to the STE embedding to bring synonymous embeddings closer while also preserving image-text similarities. The alignment transform is learned from a dataset of nearly one million pairs of corresponding textual embeddings formed from VLM and STE spaces.

The resulting VLM embedding, called SemCLIP has the desirable property that embeddings of semantically close terms and their associated images are placed close together to ensure stability in retrieval. Results of analysis and comparison on multiple benchmark datasets is indicating improved stability and quality of retrieval in comparison to other CLIP variants.

Table 1: Illustration of the semantic understanding problems in textual and VLM embeddings.

| Dissimilar sentences (Case where they should be similar) | | | Similar sentences (case where they shouldn't be similar) | | |
|---|---|---|---|---|---|
| | SBERT Score | CLIP score | | SBERT score | CLIP score |
| Orbital module | | | foxelli neoprene chest waders - camo fishing waders for men with boots | | |
| Space capsule | 0.661 | 0.846 | foxelli neoprene chest waders - material fishing waders for men with boots | 0.975 | 0.950 |
| Spacecraft | 0.621 | 0.848 | foxelli neoprene chest waders - camo commercial enterprise waders for men with boots | 0.913 | 0.980 |
| Space ship | 0.579 | 0.820 | oxelli neoprene chest waders - camo business waders for men with boots | 0.909 | 0.983 |

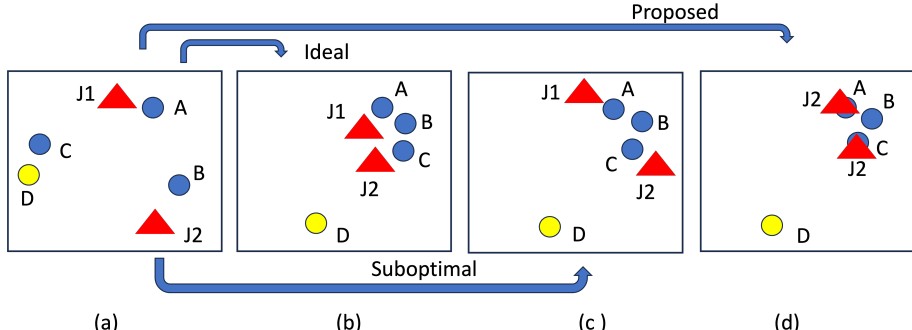

Figure 2: Illustration of the key idea proposed in this paper on achieving semantic stability through alignment transform mapping a VLM model to an STE embedding. A, B, C are textual embedding derived from 3 synonyms in original CLIP space. D is textual embedding of a non-synonym. J1 and J2 are image embeddings that project close to the text embedding A and B respectively. Goal of the alignment transform is to project all synonyms (A,B,C) and their matched images (J1,J2) close together in the target space as shown in (b). If we use text alignment alone, we can get to configuration shown in (c). If we apply the alignment transform to both text and image embedding, we get configuration shown in (d) which is close to the desired configuration in (b).

## 2 RELATED WORK

To our knowledge, insights into stability aspects of retrieval or the limitations of textual embeddings in influencing VLM models haven't been addressed in detail before. Other prior works, however, have pointed to issues with text to image retrieval and image tagging with CLIP. NegCLIP (Mert Yuksekgonul & Zou, 2023) pointed to semantic inconsistencies in responses to text queries using CLIP. It addressed the problem by fine-tuning CLIP using a dataset with hard negatives drawn from the COCO dataset by swapping different linguistic elements of the original caption. KnowledgeCLIP (Pan et al., 2022) also argued for augmenting CLIP training with knowledge graphs to allow a better understanding of the semantics in queries. StructureCLIP (Huang et al., 2023) mentions that existing methods often perform poorly on image-text matching tasks that require a detailed semantic understanding of the text and recommended augmenting VLMs with scene graphs composed of objects, attributes, and relations. BLIP (Li et al., 2022) and its variants are unified vision-language models using a multimodal mixture of encoder-decoder architectures trained with a language modeling loss to generate better captions given images. Sigmoid Loss for Language-Image Pre-training (SigLIP) (Zhai et al., 2023) introduces a pairwise sigmoid loss allowing the method to solely focus on the individual image-text pairs. Unlike CLIP which requires managing global pairwise comparison in contrastive loss, the sigmoid based loss makes the training process more scalable and flexible. The need for modeling coarse and fine-grained concepts was also emphasized in a recent work (Xu et al., 2024).

Existing CLIP variants, while offering fixes for many problems, continue to use textual embeddings derived from transformer models. In our approach, we achieve the desired improvements *by focusing at a different end, namely, improving the semantics in textual embedding* and using an alignment transform to project from the original CLIP model to form a new space of semantically connected words and images.

## 3    OVERALL APPROACH

Our goal is to transform a pre-trained joint image-text embeddings such as CLIP into a new space where the textual phrases and image pairings of synonymous words are close to each other. We use a simple example in Figure 2 to illustrate the key idea. Consider a scenario illustrated in Figure 2(a) depicting 3 synonymous terms, $A, B, C$ such as "orbital module", "space capsule" and "spacecraft" (denoted by blue circles) amidst other textual embeddings such as glass (denoted by yellow circle) in a VLM space. Let $J1, J2$ be two images that projected respectively close to $A$ and $B$, possibly reflecting space craft content (denoted by red triangles). Since the VLM textual embeddings are not necessarily capturing linguistic similarity of synonyms, these embeddings are spread out as shown in Figure 2(a). Our goal is to generate a new VLM space such as the one shown in Figure 2b, where embeddings of words and their associated images $A, B, C, J1, J2$ are all close together.

To show that the scenario shown in Figure 2(a) is likely, we conducted an experiment in which we explored the fraction of the English language nouns that are near their synonyms in existing text and VLM embeddings. Using all the over 70,000 single sense nouns from the Wordnet the-saurus (Fellbaum, 2012) and recording their top 10 neighbors in the respective embeddings using cosine similarity, we found only around 50% overlap as shown in Table 2 for both textual (SBERT) and joint image-text embeddings (CLIP).

Modeling these desired transformations more formally, let $f_i(\cdot) : X_{\text{image}} \rightarrow R^{d_i}$ be the image encoder and $f_t(\cdot) : X_{\text{text}} \rightarrow R^{d_t}$ be the text encoder. Given a batch of $N$ images, $\{I_1, I_2, ..., I_N\}$ and $N$ captions, $\{T_1, T_2, ..., T_N\}$, a VLM model projects them into a common vector space $C_i \in R^d$, $C_t \in R^d$ learned in a contrastive manner. In our notation, $C_i, C_t$ denote the vector representation in the VLM space for an image $I_i$, and a caption $T_t$, respectively. Consider two textual queries $q_1$ and $q_2$ which are synonymous of a query word $q$ (for e.g. "kreel", and "hamper" to "basket"). Let their projected vectors in the VLM space be $C_{q_1}, C_{q_2}$ and their nearest images be denoted by the vectors $C_{i_{q_1}}$ and $C_{i_{q_2}}$, respectively. Our goal is to design a new joint embedding model called SemCLIP $C^{'}$ such that:

$$|C^{'}_j - C^{'}_k| < \delta, \text{where the indexes } j, k \in \{q, q_1, q_2, i_{q_1}, i_{q_2}\} \tag{1}$$

and $\delta$ is a small neighborhood so that both images corresponding to the vectors $C_{i_{q_1}}$ and $C_{i_{q_2}}$ are pulled up to either query $q_1$ or $q_2$.

To achieve this, we use a two-stage approach. We first develop a semantic textual embedding (STE) $STE(\cdot)$, which maps individual words $W_i$ to embeddings of words $STE(W_i)$ such that

$$|STE(W_j) - STE(W_k)| < min(\gamma_{W_j}, \gamma_{W_k}), W_k \in \text{Syn}(W_j) \tag{2}$$

where $W_j, W_k$ are words related by synonym relationship as defined in Wordnet, and $\text{Syn}(W_j)$ is the synonym list of word $W_j$ and $STE(W_j)$ is the semantic embedding of word $W_j$. $\gamma_{W_j}$ is the distance over which semantic similarity holds for $W_j$. Note that the distance $\gamma_{W_j}$ is a function of $W_j$, since some words have more synonyms than others.

Once such a STE has been learned, the most straightforward approach is to train a VLM embedding (from scratch) using the STE embedding paired with images. However this is difficult for two reasons. First, if this is to be done to completely cover even a single language, this would entail generating very large datasets of image-text pairs covering all possible words or captions expressed in the language. Secondly, training such large VLM models from scratch would require huge computational resources leaving a large carbon footprint. Another possible approach is to fine-tune existing VLM models using a smaller dataset of image-text pairs and the synonymous terms directly. However, such synonyms would still be encoded using current textual embeddings which don't guarantee that synonym similarity would be maintained in the vector space as shown in Table 2. Our experiments in comparing to such fine-tuned embeddings and their results described in Section 4 also validate this assumption.

Hence in our approach, we instead design a semantic alignment transform to project the VLM embeddings to the new STE embedding. Specifically, we design an alignment transform $(\Gamma_t(\cdot), \Gamma_i(\cdot))$ that projects the textual and image embeddings from VLM space such that

$$C^{'}_t = \Gamma_t(C_t) \text{ and } C^{'}_i = \Gamma_i(C_i) \tag{3}$$

where $\Gamma_i(\cdot) : R^d \rightarrow R^d$ and $\Gamma_t(\cdot) : R^d \rightarrow R^d$. $d$ is the dimension of the STE space.

Figure 3: Architecture of SemCLIP demonstrating the various stages of creating the joint text-image embedding.

The alignment transform $\Gamma_t(\cdot)$ for text can be learned separately by mapping the embeddings of all words in a language from VLM space to the STE space. However, we cannot train separately for $\Gamma_i(\cdot)$ because the STE embedding is only defined for words. Of course, a straightforward approach would be to assume that $\Gamma_i(\cdot) = \Gamma_t(\cdot)$ and simply apply the mapping to the image vectors in the VLM space. However, this may result in sub-optimal alignment where the images projected could still be farther apart as shown in Figure 2c.

A guaranteed way to ensure that the image embeddings of synonymous words in VLM space are close to the synonymous words and their associated images in the STE space is to induce a transformation based on their nearest textual neighbors. Specifically, we can express $\Gamma_i$ as

$$\Gamma_i(C_i) = \Gamma_t(C_{t_i}) \quad \text{such that} \quad t_i = \arg\min_{t'} d(C_i, C_{t'}) \tag{4}$$

where $C_{t_i}$ is the nearest text to an image vector $C_i$ in the original VLM space in terms of distance $d(.)$: the cosine distance between the image and text vectors. This results in the image vectors aligning directly on top of the textual embedding vectors as shown in Figure 2d, thus achieving a transform closer to our original goal indicated through Figure 2b.

### 3.1 LEARNING THE SEMANTIC TEXT EMBEDDING

To derive a semantic text embedding, we use the Wordnet (Fellbaum, 2012) thesaurus where linguists have already curated related terms and defined synonyms, generalizations and specializations of concepts. To allow both meaning and sense to be captured, we adopt the lemma notation of Wordnet for representing a vocabulary word $W_i$ as:

$$W_i = <w_i, p_i, s_i, l_i> \tag{5}$$

where $w_i$ is the multi-term word, $l_i \in \text{Syn}(w_i)$ is a synonym, and $p_i \in \{n, a, v, r, s\}$ which stand for noun, adjective, verb, adverb, and adjective respectively. Finally, $s_i$ stands for the sense of the word and is a number from 1 to $n$. The advantages of the notation in capturing the sense context in detail are further elaborated in Appendix A.2.1.

**Development of a similarity list dataset**: To train our STE embedding, we generated a ground truth dataset of all groups of synonymous nouns in Wordnet. Modeling nouns alone can give sufficient coverage of vocabulary as most applications involve searching for objects denoted by nouns (Appendix A.1 presents further experimental evidence in this regard). The initial similarity lists for training were obtained by directly traversing the Wordnet ontological tree gathering synonyms (called lemmas in Wordnet) as well as hypernyms (generalizations) using the WU-Palmer similarity metric (Wu & Palmer, 1994) which is given by:

$$sim(W_i, W_j) = 2 * \text{depth}[\text{lcs}(W_i, W_j)]/[\text{depth}(W_i) + \text{depth}(W_j)] \tag{6}$$

where where $\text{lcs}(W_i, W_j)$ is the least common ancestor of $W_i$ and $W_j$ and depth$(\cdot)$ stands for the depth of the concept in the ontology.

Without a constraint on the depth differential (2 in our case), and a reasonably high threshold, the WUP similarity score alone can reveal several false positives in association and lead to undesirable wider expansion of meanings, particularly for words closer to the root of the WordNet hierarchy. For example, with a 4 level depth differential for a word such as 'chair.n.05.chair' , the WUP similarity to the word 'device.n.01.device' is high (0.823) which is not synonymous. Therefore, to normalize the notion of similarity, the initial lists produced by the automatic algorithm were curated by domain specialists. For Wordnet, we used a team of 3 linguists from a nearby university to examine the similarity lists so that relationships other than similarity in meaning and sense were removed. Each linguists produced their own curated similarity lists. Triple consensus process was used to filter the lists so that those terms identified by all 3 linguists were retained in the final similarity list per anchor words. The original scores returned by the WUP metric were still retained for these pairs so that the linguists only filtered the irrelevant words from the lists but did not alter the WUP scores. For the Wordnet ontology, we were able to address all valid nouns and their synonyms resulting in over 140,000 words. *Note that this vocabulary already exceeds the token vocabulary of most transformer models.* The whole curation process took over 1 year to complete. Table 6 in the appendix captures more details of this painstaking process. Appendix A2.2 gives further details on our rationale and the workflow used to construct the similarity lists.

**Building the Semantic textual embedding (STE) model:** We developed a contrastive embedding model that captures the essence of the similarity in the curated similarity lists in a numerical formulation. The unique words extracted from the similarity lists $Syn(W_i)$ for each anchor term $W_i$ form the base vocabulary for our embedding. Pairs of anchor and target words from similarity lists are taken as positive examples. All other pairings represent negative examples for the anchor class. The word embedding was then learned using multi-class supervised contrastive learning (Khosla et al., 2020).

Given a fully-specified 4-tuple anchor word $W_i$, we encode it by a 1-hot encoding $O_i \in \{0, 1\}^{|V|}$, s.t. $\sum_{j=1}^{|V|} O_{ij} = 1$ as an input to the network where $V$ is the vocabulary. As a supervision label, we form a label vector in the real number space $Y_i = R^{|V|}$, s.t. $Y_{ij} = sim(W_i, W_j)$ iff $W_j \in Syn(W_i)$ and 0 otherwise. Here $sim(W_i, W_j)$ is the similarity score returned from the similarity list generation. Thus each similarity list is characterized by a unique pattern label vector.

We generate a new encoder-decoder network consisting of an embedding layer (ker, 2021) to handle the large one hot vectors, a dense fully connected layer with ReLU activation for an encoder, and a decoder/projection network as another fully connected layer with ReLU activation as shown in Figure 3 (right side). The similarity between an anchor word $W_i$ at index $i$ in the vocabulary $\mathcal{V}$, and a candidate word $W_j \in Syn(W_i)$ is captured by the contrastive loss per similarity list as:

$$\ell_{\text{contrast}}(S_i) = - \sum_{W_j \in Syn(W_i)} \log \frac{\exp(z_i \cdot z_j / \tau)}{\sum_{a \in V} \exp(z_i \cdot z_a / \tau)} \tag{7}$$

Here $z_i$ is the projected vector for word $W_i$ and $z_j$ is the projected vector similarly for $W_j \in Syn(W_i)$. Finally, $z_a$ is the projected vector for any word $W_a$ either inside or outside the similarity list (i.e. ideally the entire vocabulary). $\tau$ is the temperature to weigh the contribution from similar vectors. Also, since there are multiple such similarity lists, one for each vocabulary term, we can train them in sequential fashion through batching using a cumulative contrastive loss as $\mathcal{L}_{contrast} = \sum_{j}^{|\mathcal{V}|} \ell_{contrast}(S_j)$

**Implementation Details:** Overall, the designed network architecture had the following parameters: input and output vector sizes$= 142, 989$, for various encoding size $= 300, 1024, 2048, 4096$, and temperature$= 0.05$ in the loss function. We used a batch size of $800$ and trained over a maximum of $10$ epochs or until the network error convergence was reached. We used the Adam optimizer for fast convergence with the learning rate as $0.001$. Two NVIDIA P100 GPUs with 16 GB were used for training and training took 5 hours. The network overall had 43,666,800 parameters (for encoding size of 300) and scaled accordingly for higher size encodings.

## 3.2 Learning the alignment transform

To learn the alignment transform, we form a ground truth dataset of pairs of embeddings derived from a VLM model and the STE model for candidate words or phrases. While this method could be applied to any joint image-text embedding VLM model, the alignment model was developed for the original CLIP model Radford et al. (2021).

**Forming an alignment dataset:** Unlike the STE embedding which was derived from WordNet, the alignment mapping used additionally, a much larger vocabulary of captions. Table 9 in the Appendix provides more details on the nearly 800,000 captions accumulated across datasets that were used to train the alignment transform. For this, the corresponding pairs of entities needed to be defined between the source VLM embedding $C_q$ and the target semantic embedding $C_q'$. For the candidate words in Wordnet, to preserve the word sense in the correspondence, we formed pairs of words with their synonyms as defined directly in Wordnet. For example, to capture the correspondence for 'seal as in the animal seal', we denote the correspondence by the pair (seal.n.09.seal $\rightarrow$ 'seal as in sea animal') where the phrase after 'as in ' is directly derived from the synonym descriptions supplied in Wordnet. This ensures uniqueness of correspondence between the STE and VLM embeddings.

For long captions, the correspondence was derived from the composed words in the caption and forming their average embeddings. Our experiments indicated that the VLM models are relatively robust to approximating embeddings of captions by the average embedding formed from their constituent nouns. Section A.1 in Appendix presents a detailed rationale for the use of average vectors along with further results from VLM experiments to support this observation.

The captions were spelling corrected before using their composed words for averaging. As for out-of-vocabulary words, we found the nearest match to their lexical variants in the vocabulary using an SBERTReimers & Gurevych (2019b) encoding of the words/phrases. Since the nouns in the captions could be associated with multiple senses, an available word sense disambiguation (WSD) tool, ESC (Barba et al., 2021), was employed to resolve the sense of the constituent nouns before making the correspondence. More details on word-sense disambiguation are available in Appendix A.3.2.

**Implementation of the alignment transform** The alignment transform $\Gamma_t(\cdot)$ is a three layered Multi-layered Perceptron (MLP) with input size 512, output size 300 and intermediate layer width 4096 as shown in Figure 3 (middle). We use Layer Norm as the activation function. The network is trained using a Mean Squared Error (MSE) loss between the neural network outputs and the ground truth semantic embeddings. Equation 8 below captures the network details.

$$\underline{\text{Transform:}} \ \Gamma_t(\cdot) = \mathbf{FC}_3(\mathbf{\Phi}_{\text{relu}}(\mathbf{FC}_2(\mathbf{\Phi}_{\text{relu}}(\mathbf{FC}_1((\cdot)))))) \ \underline{\text{Loss:}} \ \mathcal{L} = ||\Gamma_t(C_t) - \mathbf{C}_t'||_2^2 \quad (8)$$

To train the network, we use the ADAM optimizer with weight regularization (AdamW) and initial learning rate as 0.001. We train for a total of 200 epochs and use a batch size of 512. Along with the decrease in training loss, we calculate the retrieval errors (i.e. training fit using a nearest neighbors matching) after projection and observe less than 4 percent error in recovering the target semantic embeddings post-projection after training. Once $\Gamma_t(\cdot)$ is learned, the images were mapped to the nearest text vector and projected using the learned alignment transform as described in Section 3 and as shown by the cross-links between image and text embeddings in the overall end-to-end architecture of SemCLIP in Figure 3 (left).

## 4 Results

The SemCLIP model and its constituent embeddings were evaluated for semantic stability of image retrieval on a variety of datasets as this was the prime objective of developing this embedding. However, we also conducted extensive studies documenting the performance of SemCLIP for many relevant downstream tasks such as image-to-text, text-to-image retrieval, and text-to-text retrieval for the STE embedding as well.

**Datasets**: We compare the performance of STE embedding on 13 benchmark datasets as listed in Table 3. All datasets contain pairs of terms that are related in multiple ways ranging from synonyms to antonyms, to part-of relations and have been used in previous previous evaluations. For the joint embedding, we evaluated the performance of SemCLIP on 5 datasets, namely, Visual Genome (Krishna et al., 2016), SUN (Xiao et al., 2010), CUB (Wah et al., 2023), AWA2 (Xian et al., 2019),

Table 2: Illustration of synonym recognition across text embeddings.

| Embedding | # Queries | Synonyms in Top10 | %age synonyms covered |
|---|---|---|---|
| CLIP (Radford et al., 2021) | 71895 | 28070 | 49.27% |
| SBERT (Reimers & Gurevych, 2019b) | 71895 | 37888 | 52.7% |
| Ours | 71895 | 67309 | 87.7% |

Table 3: Illustration of comparative performance of semantic textual embeddings (STE) on benchmark datasets. The last column shows the STE result for the similar subset.

| Datasets | Original Word # | WordNet Filtered | Word2Vec | Glove | BERT | Path2Vec | STE | STE |
|---|---|---|---|---|---|---|---|---|
| EM_SIMLEX_SYNS | 297 | 297 | 0.285 | 0.240 | 0.145 | 0.301 | 0.265 | **0.570** |
| EN-MC-30 | 30 | 30 | **0.789** | 0.702 | 0.410 | **0.782** | 0.650 | 0.650 |
| EN-MEN-TR-3k | 3000 | 2657 | **0.776** | 0.743 | 0.310 | 0.366 | 0.257 | **0.780** |
| EN-MTurk-287 | 287 | 243 | 0.767 | 0.705 | 0.435 | 0.317 | 0.300 | **0.810** |
| EN-MTurk-771 | 771 | 771 | 0.671 | 0.649 | 0.335 | 0.404 | 0.466 | **0.760** |
| EN-RG-65 | 65 | 64 | 0.761 | 0.770 | 0.446 | 0.723 | 0.640 | **0.820** |
| EN-RW-STANFORD | 2034 | 910 | 0.492 | 0.341 | 0.226 | 0.194 | 0.217 | **0.590** |
| EN-SIMLEX | 666 | 666 | 0.452 | 0.397 | 0.233 | 0.505 | 0.398 | **0.670** |
| EN-WS-353-REL | 252 | 248 | **0.626** | 0.578 | 0.159 | 0.136 | 0 | 0 |
| EN-WS-353-SIM | 203 | 201 | 0.774 | 0.659 | 0.388 | 0.599 | **0.820** | **0.820** |
| EN-YP-130 | 130 | 43 | 0.542 | 0.545 | 0.326 | 0.029 | 0.426 | **0.660** |
| EW-WS-353-Syns | 99 | 98 | 0.507 | 0.507 | 0.366 | 0.616 | **0.655** | **0.655** |
| EN-WS-353-ALL | 352 | 348 | 0.694 | 0.607 | 0.256 | 0.406 | 0.303 | **0.720** |

MS-COCO, and Flicker30k. In each case, we retained all the labels and the test image partition provided for these datasets. Each of the labels was processed using Spacy to extract all noun entities. We then resolved their sense to give a 4-part notation for the nouns as described earlier. The details of these datasets are described in Table 5 and Table 4.

**Comparison methods**: The semantic text embedding was compared to 4 popular word embedding methods including, Word2Vec, Glove, BERT (Devlin et al., 2018), and Path2Vec (Kutuzov et al., 2019). Since most image-text embeddings are variants of CLIP (Radford et al., 2021), our comparisons for SemCLIP included all popular variants, namely, Open AI's original CLIP (Radford et al., 2021), OpenCLIP (Radford et al., 2021), NegCLIP (Mert Yuksekgonul & Zou, 2023), and BLIP (Li et al., 2022). In addition, we conducted ablation studies creating a variant of CLIP called PosCLIP by fine-tuning CLIP directly with synonymous captions. More details are provided in Appendix A.4 and Table 10.

**Recognition of synonyms through STE model**: Since the STE model was trained with synonym similarity lists, we expect a high overlap with synonyms in its topK retrieval in comparison to other textual and VLM embeddings. To record this, we repeated the experiment described in Section 3 using SemCLIP embedding and the result is shown as the last row in Table 2 indicating *nearly a doubling of performance* over popular existing embeddings. Qualitatively, we found the similarity lists produced by STE embedding to consists of only synonymous terms unlike other encodings like Word2Vec. Appendix A.2.4 has further details on our observations captured in Table 8. Also, the list of similar terms found by searching in STE embedding is larger than the pure list of synonyms found in Wordnet due to the learning process that infers more related terms. Again, we refer to Appendix A.2.3 and its Table 7 for further discussion.

**Quantitative evaluation of STE embedding:** For quantitative results, we evaluated the performance of STE embedding on 13 textual benchmarks shown in Table 3. The resulting performance using the Spearman correlation coefficient to see the agreement of the similarity ranked lists produced for each word in comparison to human ranked lists, is shown in that table. Our method was expected to perform worse on the datasets where the relations are antonyms or other forms of relations besides meaning similarity, but should perform better when limited to the meaning-wise similar pairs in these benchmark datasets. As seen in Table 3, it significantly outperforms other embeddings in the case of the EN-WS-353-SIM dataset which focuses on similarity relations. If we restrict the analysis to only the similar words in all datasets, our method outperforms all other methods as shown in the last column. Finally, for datasets such as EN-WS-353-REL which capture antonyms and other relationships besides synonyms, our performance is the least, which is also a good result indicating it is able to focus on similarity relations only. Note that the values in Table 3

Table 4: Results of average text-image retrieval overlap when querying using synonyms of nouns in the respective datasets. For each query, we use ten synonyms to estimate the image retrieval overlap. The CLIP-G stands for CLIP-ViT-BigG model.

| Dataset | Images/Queries | Method | Overlap@1 | Overlap@5 | Overlap@10 | Overlap@50 |
|---|---|---|---|---|---|---|
| Visual Genome | 7794 / 14513 | **SemCLIP** | **0.551** | **0.532** | **0.517** | **0.523** |
| | | CLIP-G | 0.245 | 0.245 | 0.246 | 0.246 |
| | | CLIP | 0.119 | 0.056 | 0.038 | 0.026 |
| | | OpenCLIP | 0.129 | 0.055 | 0.040 | 0.025 |
| | | BLIP | 0.119 | 0.056 | 0.037 | 0.024 |
| | | NegCLIP | 0.109 | 0.063 | 0.038 | 0.024 |
| CUB | 11788 / 200 | **SemCLIP** | **0.812** | **0.783** | **0.732** | **0.715** |
| | | CLIP-G | 0.127 | 0.143 | 0.170 | 0.218 |
| | | CLIP | 0.118 | 0.072 | 0.062 | 0.053 |
| | | OpenCLIP | 0.149 | 0.079 | 0.062 | 0.053 |
| | | BLIP | 0.181 | 0.084 | 0.066 | 0.053 |
| | | NegCLIP | 0.119 | 0.078 | 0.066 | 0.055 |
| SUN | 16657 / 567 | **SemCLIP** | **0.554** | **0.531** | **0.523** | **0.511** |
| | | CLIP-G | 0.214 | 0.216 | 0.221 | 0.225 |
| | | CLIP | 0.092 | 0.048 | 0.034 | 0.025 |
| | | OpenCLIP | 0.070 | 0.039 | 0.028 | 0.021 |
| | | BLIP | 0.091 | 0.05 | 0.035 | 0.025 |
| | | NegCLIP | 0.095 | 0.045 | 0.032 | 0.023 |
| AWA2 | 6985 / 10 | **SemCLIP** | **0.751** | **0.723** | **0.702** | **0.715** |
| | | CLIP-G | 0.110 | 0.184 | 0.186 | 0.246 |
| | | CLIP | 0.086 | 0.051 | 0.038 | 0.028 |
| | | OpenCLIP | 0.139 | 0.067 | 0.046 | 0.032 |
| | | BLIP | 0.139 | 0.057 | 0.039 | 0.028 |
| | | NegCLIP | 0.101 | 0.046 | 0.034 | 0.025 |

are Spearman correlation coefficient where the values above 0.7 indicate strong correlation which our method achieves for most datasets.

**Evaluating the stability of text-to-image retrieval**: We evaluated the stability of retrieval by measuring the overlap in the image lists returned in response to queries and their synonym variants. Specifically, we extracted nouns from each of the captions covered by the test partitions of the respective datasets. All text to image retrieval used a common prompt of "A photo of " before each noun flagged in a caption. We then recorded the pairwise overlap of the top K lists returned for a caption with the top K lists of images returned from their synonym replacements. The overlap was averaged across the synonym replacements to serve as a measure of the stability of retrieval. The experiments were performed for all CLIP variants including a newly released much larger backbone CLIP ViT-bigG. The result is shown in Table 4. As can be seen, by projecting the synonymous phrases to the SemCLIP embedding, the list of images returned show far higher overlap in SemCLIP in comparison to other CLIP variants.

**Evaluating text-to-image retrieval**: Due to the projection of synonymous phrases and their associated embedding close together, we expect an increase in the precision and recall for general text-to-image retrieval as well. We evaluated this using the popular measures of NDCG and mean average precision (MAP). To keep the comparison fair, all ground truth labels of images were augmented with their synonym equivalents. For example, images labeled with 'clock frame' were also augmented with the label 'frame/clock' from the same caption set as both these labels share the same entities and would be represented by the same average vector in SemCLIP space. For each dataset, our method achieves the highest NDCG@K as well as MAP across various datasets as shown in Table 5 (under the columns "t2i") except for AWA2 which had the fewest labels.

**Evaluating image-to-text retrieval:** The image-to-text retrieval experiments results also showed similar performance as shown in Table 5 under the columns "i2t". Note that when there are large number of captions (visual genome, Flickr30k), our method's performance is best seen due to the capturing of semantics of multiple noun phrases in the average vector embeddings used in the transformation. The COCO and Flickr30K labels were not used for training the alignment mapping of SemCLIP.

**Performance on classification:** We evaluated SemCLIP also for the standard task of classification using the predicted labels. On the ImageNet classification, our zero shot classification accuracy for SemCLIP was at 88.3% in comparison to CLIP at 84.2%.

Table 5: Comparisons of text-to-image (t2i) and image-to-text (i2t) retrieval performance with different models.

| Dataset | Images / Labels | Model | NDCG@10 (t2i) | mAP@10 (t2i) | NDCG@10 (i2t) | mAP@10 (i2t) |
|---------|-----------------|-------|---------------|--------------|---------------|---------------|
| **Visual Genome** | 7794 / 14513 | **SemCLIP** | **0.192** | **0.172** | **0.254** | **0.185** |
| | | CLIP | 0.053 | 0.060 | 0.050 | 0.129 |
| | | OpenCLIP | 0.066 | 0.075 | 0.061 | 0.159 |
| | | BLIP | 0.072 | 0.081 | 0.069 | 0.167 |
| | | NegCLIP | 0.063 | 0.072 | 0.060 | 0.150 |
| | | PosCLIP | 0.074 | 0.084 | 0.060 | 0.146 |
| **CUB** | 11788 / 200 | **SemCLIP** | **0.721** | **0.845** | **0.891** | **0.812** |
| | | CLIP | 0.513 | 0.621 | 0.619 | 0.554 |
| | | OpenCLIP | 0.669 | 0.744 | 0.777 | 0.726 |
| | | BLIP | 0.204 | 0.341 | 0.326 | 0.260 |
| | | NegCLIP | 0.406 | 0.535 | 0.472 | 0.403 |
| | | PosCLIP | 0.488 | 0.580 | 0.553 | 0.479 |
| **SUN** | 16657 / 567 | **SemCLIP** | **0.686** | **0.712** | **0.810** | **0.671** |
| | | CLIP | 0.414 | 0.562 | 0.458 | 0.415 |
| | | OpenCLIP | 0.549 | 0.664 | 0.514 | 0.476 |
| | | BLIP | 0.413 | 0.535 | 0.426 | 0.384 |
| | | NegCLIP | 0.429 | 0.553 | 0.425 | 0.380 |
| | | PosCLIP | 0.463 | 0.602 | 0.445 | 0.399 |
| **AWA2** | 6985 / 10 | **SemCLIP** | 0.967 | 0.987 | 0.995 | 0.989 |
| | | CLIP | 0.993 | 0.999 | 0.992 | 0.989 |
| | | OpenCLIP | **1.000** | **1.000** | **0.994** | **0.991** |
| | | BLIP | **1.000** | **1.000** | 0.991 | 0.988 |
| | | NegCLIP | **1.000** | **1.000** | 0.987 | 0.982 |
| | | PosCLIP | **1.000** | **1.000** | 0.983 | 0.977 |
| **COCO** | 5000 / 80 | **SemCLIP** | **0.940** | 0.91 | **0.895** | 0.923 |
| | | CLIP | 0.810 | 0.869 | 0.706 | 0.771 |
| | | OpenCLIP | 0.866 | 0.926 | 0.756 | 0.788 |
| | | BLIP | 0.897 | 0.942 | 0.774 | 0.852 |
| | | NegCLIP | 0.834 | 0.892 | 0.766 | 0.797 |
| | | PosCLIP | 0.919 | 0.946 | 0.807 | 0.834 |
| **Flickr30k** | 31014 / 158391 | **SemCLIP** | **0.571** | **0.523** | **0.580** | **0.572** |
| | | CLIP | 0.354 | 0.306 | 0.314 | 0.430 |
| | | OpenCLIP | 0.427 | 0.377 | 0.376 | 0.489 |
| | | BLIP | 0.562 | 0.510 | 0.475 | 0.587 |
| | | NegCLIP | 0.425 | 0.373 | 0.354 | 0.465 |
| | | PosCLIP | 0.323 | 0.279 | 0.273 | 0.367 |

**Ablation studies**: We conducted an ablation study in which we fine-tuned CLIP on the visual genome dataset using synonymous captions. A variant of CLIP called PosCLIP was developed, which was fine-tuned on the Visual Genome dataset. In PosCLIP, hard positives were used with binary cross-entropy loss. To construct hard positives and hard negatives given a caption, the captions were processed to extract entity nouns. Each noun was then replaced by its synonym from Wordnet to form caption variants. The details of this processing are available in Appendix A.4. As seen in Table 5, for fine-tuned variants of CLIP (POSCLIP), the results were not as impressive as doing an explicit transformation of the terms into the SemCLIP space, indicating the dominance of prior large-scale training of the underlying models.

**Discussion:** We note from Table 4 that the performance of retrieval and retrieval overlap varies among datasets (0.21 to 0.72 and 0.55 to 0.81 respectively). This is both due to nature of the labels as well as their number. It is also due to incomplete ground truth labeling for datasets such as Visual Genome which have larger vocabularies. Overall, the performance of SemCLIP was better for larger vocabularies due to the better handling of synonymous terms, and it was similar to others for datasets with smaller vocabularies.

## 5 CONCLUSIONS AND LIMITATIONS

In this paper, we offered new insights into VLM models in terms of modeling synonymous relationships and proposed new approaches based on alignment transform to a newly created semantic textual embedding. The semantic text embedding was developed only for nouns in the English language. In designing the transformation mapping, a sense disambiguation tool was used whose accuracy is also known to be limited (around 80%).

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

## A  APPENDIX

In this appendix, we present additional details to support the conclusions of our paper. The paper itself is self-contained and covered many aspects of our SemCLIP model generation. However, there were several additional experiments done that can throw some more light into the work done for the paper. These are captured in the sections below.

### A.1  HANDLING ARBITRARY TEXT CAPTIONS THROUGH AVERAGE VECTORS

In this section, we present the rationale for representing arbitrary captions through the average embedding formed from their composed words. The composed words are derived from the Wordnet thesaurus. The following discussion is applicable other textual embeddings besides our STE model. The notations used here are the same as in Section 3.

The rationale for the average vector approach comes from two sources. First, the VLMs are able to retrieve relevant images to textual queries even when they are expressed simply as a collection of grammatical entities. Consider a full caption: "There is a table in the middle of the room". The composed non-stop and useful words in this sentence can be easily extracted through standard NLP methods as "table", "middle", "room". If we ignore the preposition, and focus only on nouns, then the composed nouns are "table" and "room". By using the average CLIP text vectors of these nouns, the images retrieved are roughly similar to those retrieved by the use of the full caption as seen from Figure 1(e)-(g) where relevant matches are obtained from both a full fledged phrase shown (Figure 1(e)) as well as when broken down into a set of nouns only ("table", "room") (Figure 1(f)) or in any order ("room", "table") in Figure 1(g). Thus it seems plausible to represent an arbitrary caption in terms of its essential composed words and in particular, the constituent nouns depicting objects.

Our experimental validation also showed that replacing queries by the average vectors of embeddings from their composed nouns gives similar retrieval performance. Figure 4(a) and (b) shows the results of finding similar captions in the CLIP embedding space for the entire set of 83404 captions in the Visual Genome dataset (Krishna et al., 2016) based on their average vectors. As can be seen from Figure 4(a), the original caption was the nearest vector for 90% of the average vectors with an average cosine similarity of 0.958. We also repeated this in an image to text similarity experiment using the original captions vectors and their average versions on all the 7554 images of the test partition of the Visual Genome dataset (Krishna et al., 2016). As can be seen from the results in Table in Figure 4(b), the performance using average vectors is comparable to the performance with the original captions.

In fact, we can make the following proposition.

**Proposition-1:** The vector representation $C_q$ of a query $Q$ in a VLM space $C$ can be approximated by the average vector $C_{\text{avg}} = \frac{\sum_j C_{ej}}{N_q}$ where $C_{ej}$ is the vector representation of the entity $e_j$ in the VLM space $C$ and $N_q$ are the number of entities composing the query $Q$.

A second rationale comes from the fact that if we develop a semantic text embedding for words that preserve the synonymous relationship of individual grammatical elements, e.g. nouns, we can expect their enclosing synonymous queries to preserve their relationships in the projected space as well.

**Corollary-1:** Given pairs of synonymous queries $Q_1, Q_2$ represented by their average vectors $C'_{\text{avg1}}, C'_{\text{avg2}}$ formed from $C'_{eq11}, .. C'_{eq1k}$ and $C'_{eq21}, .. C'_{eq2k}$, if $|C'_{eq1l} - C'_{eq2l}| < \delta$ for all $C'_{eql}$ then $|C'_{\text{avg1}} - C'_{\text{avg2}}| < \delta$. This follows directly from vector averaging rules.

### A.2  MORE DETAILS ON SEMANTIC TEXT EMBEDDING LEARNING

In this section, we provide further details on our semantic text embedding.

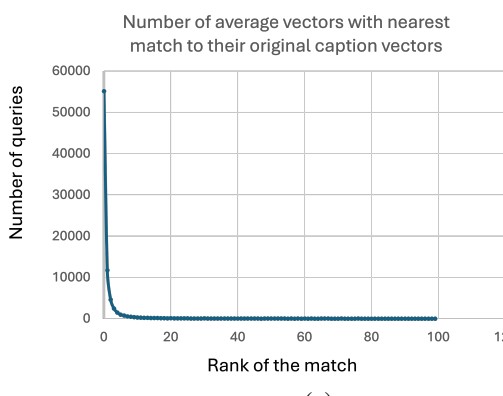

| Top K | %match using full caption | % match using average vector |
|---|---|---|
| 1 | 4.38% | 3.34% |
| 5 | 11.70% | 10.26% |
| 10 | 16.35% | 14.90% |
| 50 | 32.04% | 30.52% |

(a)             (b)

Figure 4: (a) Illustration of text-to-text retrieval using captions approximated by average vectors. (b) Illustration of image-to-text retrieval using full caption vectors and approximation by average vectors.

### A.2.1 REPRESENTING WORD SENSES IN STE EMBEDDING

We note that unlike other word embeddings which either have a unique embedding (e.g. Word2Vec) or variable embeddings based on use context (e.g. BERT), our representations of a word in STE embedding are only as many as the senses in which the word occurs in the language. Consider an example word 'lemon' which has 5 senses, even though not all 5 of them begin with the word lemon in the synset definitions of Wordnet. Lemon is a lemma (in Wordnet, the synonyms are captured as lemmas) in 'lemon.n.01.lemon' using the synset 'lemon.n.01' which has the meaning "'yellow oval fruit with juicy acidic flesh' . Lemon is also a lemma or synonym of the word 'gamboge' in the form 'gamboge.n.01.lemon' whose synset 'gamboge.n.01' has the meaning 'a gum resin used as a yellow pigment and a purgative' so that the reference to lemon here is for its color. In our STE embedding, 'lemon.n.01.lemon' and 'gamboge.n.01.lemon' are two different embeddings and will have two different similarity sets, the former grouping lemon variants of the fruit, while the latter referring to resins and gums.

### A.2.2 RATIONALE FOR CURATION OF SIMILARITY LISTS BY LINGUISTS

While the WUP metric can give an initial indication of word similarity, this alone is insufficient. Without a constraint on the depth differential (2 in our case), and a reasonably high threshold, the WUP similarity score alone can reveal several false positives in association and lead to undesirable wider expansion of meanings, particularly for words closer to the root of the WordNet hierarchy. For example, with a 4 level depth differential for a word such as 'chair.n.05.chair' , the WUP similarity to the word 'device.n.01.device' is high (0.823) which is not synonymous. Further, the metric does not give a complete picture of semantic distance since domain-specific ontologies were constructed before their planned uses in textual embeddings. Also, due to the nature of the English language, the shortest-path distances between nodes or ontological depth differences do not have a uniform implication of similarity across words. For example, synsets 'car.n.01' and 'van.n.01' are 16 apart in shortest path length, while 'car.n.01' and 'automobile.n.01' are only 1 apart. Conversely, terms that are not so close in meaning could also end up having a high score. Hence our approach was to filter such initial similarity lists with manual validation by linguists.

The overall workflow of the curation process and its use in training the STE embedding is illustrated in Figure 5. As can be seen, all operations except the verification by linguists/domain-specialists are done automatically. The curation process used for cleaning the similarity lists automatically traversed in Wordnet removed many of the spurious similarities. Table 6 shows an example for one such similarity list for the word mutual_fund.

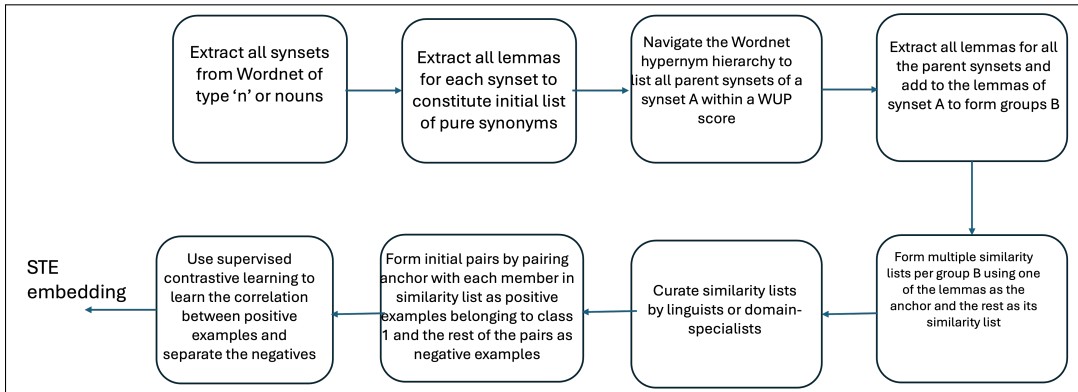

Figure 5: Illustration of the similarity lists creation and curation for STE embedding training.

Table 6: Illustration of the similarity curation process with linguists. The initial similarity list for the concept Fund.n.01.mutual_fund had over 20 nouns which were filtered to 10 by the linguists as shown in the third column.

| Initial Similarity List I | Initial similarity list II (Contd.) | Retained after curation (11) |
|---|---|---|
| budget.n.01.budget 0.94
civil_list.n.01.civil_list 0.89
demand_deposit.n.01.demand_deposit 0.89
deposit.n.04.deposit 0.94
exchange_traded_fund.n.01.exchange
_traded_fund 0.89
fund.n.01.fund 1.00
fund.n.01.monetary_fund 1.00
index_fund.n.01.index_fund 0.89
medium_of_exchange.n.01.medium
_of_exchange 0.86
money.n.01.money 0.93 | mutual_fund.n.01.mutual_fund 0.94
operating_budget.n.01.operating
_budget 0.89
pension_fund.n.01.superannuation
_fund 0.94
petty_cash.n.01.petty_cash 0.94
revolving_fund.n.01.revolving
_fund 0.94
savings.n.01.nest_egg 0.94
sinking_fund.n.01.sinking_fund 0.94
slush_fund.n.01.slush_fund 0.94
trust_fund.n.01.trust_fund 0.94
war_chest.n.01.war_chest 0.94 | exchange_traded_fund.n.01.exchange
_traded_fund 0.89
fund.n.01.fund 1.00
fund.n.01.monetary_fund 1.00
index_fund.n.01.index_fund 0.89
money.n.01.money 0.93
mutual_fund.n.01.mutual_fund 0.94
pension_fund.n.01.superannuation
_fund 0.94
revolving_fund.n.01.revolving
_fund 0.94
sinking_fund.n.01.sinking_fund 0.94
slush_fund.n.01.slush_fund 0.94
trust_fund.n.01.trust_fund 0.94 |

### A.2.3 STE EMBEDDING IS MORE THAN LOOKING UP WORDNET FOR SYNONYMS

The semantic text embedding developed covers many more synonymous relationships between nouns than explicitly captured through direct synonyms, hypernyms and hyponyms of Wordnet although they are recoverable from Wordnet through careful navigation. Table 7 lists a few examples indicating the expanded list produced by searching in our semantic text embedding.

### A.2.4 QUALITATIVE COMPARISON OF STE WITH WORD2VEC

The STE embedding by way of training with similarity lists ensures that the top matches all captures the synonymous relations in comparison to other textual encodings such as Word2Vec. This is illustrated in Table 8.

### A.3 MORE DETAILS ON ALIGNMENT TRANSFORM LEARNING

### A.3.1 CAPTIONS USED FOR TRAINING THE ALIGNMENT TRANSFORM

Table 9 records the details of the entity breakdown process used for analyzing the captions in the various datasets used for training our transformation mapping. A total of 799702 captions were used that included the WordNet nouns as well.

### A.3.2 WORD SENSE DISAMBIGUATION USED

We used the word-sense disambiguation tool ESC(Barba et al., 2021) for parsing the captions to find the correct sense of the constituent nouns in the caption. Specifically, for each noun in a caption, we used the noun as the target word and the caption as the context, and the output was the sense of the

Table 7: Illustration of synonym expansion through SemCLIP text embedding to show that the retrieved synonyms are more than what can be obtained by Wordnet alone indicating the additional value of SemCLIP for other downstream use cases requiring semantic text analysis.

| Query | Search in Wordnet | Search in Semantic Embedding |
|---|---|---|
| 'hood | {'hood', vicinity} | {'hood', 'proximity', 'gold_coast', 'locality', 'neighbourhood','neck_of_the_woods', 'neighborhood', 'place', 'section', 'vicinity'} |
| abdominal cavity | {'abdominal_cavity', 'cavity', 'abdomen'} | {'pit_of_the_stomach', 'orbital_cavity', 'glenoid_cavity', 'cavity', 'axillary_fossa', 'abdomen', 'orbit', 'cavum', 'abdominal_cavity', 'bodily_cavity'} |
| erosion | {'erosion', 'ablation'} | {'erosion', 'deflation', 'wearing_away', 'ablation', 'detrition', 'eroding', 'abrasion', 'attrition', 'eating_away', 'wearing'} |
| dorm room | {'dorm_room', 'dormitory_room', 'dormitory', 'bedroom'} | {'dormitory_room', 'chamber', 'sleeping_accommodation', 'master_bedroom', 'dormitory', 'dorm_room', 'sleeping_room', 'bedroom', 'bedchamber', 'guestroom'} |

Table 8: Sample top 10 results below show the quality of matches from Word2Vec versus our embedding where non-synonyms can be seen in the Word2Vec list.

| Query | Top 10 Results |
|---|---|
| van (Word2Vec) | 'car', 'parking lot', 'parking meter', 'friend', 'back', 'suv', 'two', 'vehicle', 'street sign', 'front' |
| tree trunk (Word2Vec) | 'tree trunk', 'trunk', 'tree', 'tree branch', 'pole', 'pine tree', 'ski pole', 'christmas tree', 'telephone pole', 'line' |
| van (STE) | 'van.n.01', 'car.n.01', 'sport utility.n.01', 'jeep.n.01', 'cab.n.01', 'minivan.n.01', 'sedan.n.01', 'automotive vehicle.n.01', 'motor vehicle.n.01', 'delivery van.n.01' |
| tree trunk (STE) | 'tree trunk.n.01', 'plant organ.n.01', 'trunk.n.01', 'stalk.n.02', 'bole.n.01', 'stem.n.02', 'wood.n.01', 'pole.n.01', 'structure.n.01' |

noun in the 4-part notation mentioned given in Eqn (5). We used the model checkpoint [1] provided by the authors of ESC to do the sense disambiguation. This model, given a target noun, and a sentence, picks the best sense of the noun in terms of Wordnet synsets. For example, in a phrase such as "Lion and giraffe in separated enclosures at the zoo", and the target noun "lion" it disambiguates among the three senses of nouns in Wordnet and correctly picks the sense 'lion.n.01' (lion as an animal) against 'lion.n.02' (celebrity) or 'lion.n.03' (leo sign of the zodiac). Among the WSD tools available in literature, this was the best performing with an accuracy of 80% indicating this is still a challenging research problem. For example, in the sentence "an old fashioned colonial dining room hutch and an anniversary clock on a shelf on the wall" and using the target noun as 'hutch', it maps to the synset 'hovel.n.01' which means a crude shelter and not the furniture as intended here.

## A.4    GENERATING FINE-TUNED CLIP MODEL USING SYNONYMS

The ablation study showed that fine-tuning CLIP using synonyms of words or creating synonymous variants of captions as positive examples does not offer the same advantages as projecting CLIP embeddings to STE embeddings. While synonyms of single word captions can be directly looked up in Wordnet, generating synonymous phrases for arbitrary captions posed challenges since not every substitution resulted in a meaningful caption. Table 10 shows this through the manipulation of

---

[1]https://github.com/SapienzaNLP/esc

Table 9: Details of captions from various datasets used for training the transformation mapping.

| Dataset | Captions | Captions with Noun | Captions with Entity | Caption with Noun Phrase | Caption with Tokens |
|---------|----------|--------------------|--------------------|------------------------|--------------------|
| MS-COCO | 568456 | 568372 | 96956 | 568414 | 568456 |
| Visual Genome | 83404 | 69410 | 14157 | 76121 | 83404 |
| CUB | 200 | 58 | 46 | 109 | 200 |
| SUN | 567 | 239 | 105 | 357 | 567 |
| AWA | 50 | 22 | 9 | 39 | 50 |
| Wordnet | | 147025 | | | |
| **Total** | | | | | 799702 |

Table 10: Illustration of positive examples generation for training the PosCLIP model. Less sensible captions generated by synonym substitutions are filtered by SBERT.

| Original phrase | pony toy |
|-----------------|----------|
| Synonyms of each word | toy → plaything, water pistol, hobby, rocking horse, slingshot, catapult |
| | pony → cayuse, Indian pony, horse, Equus caballus |
| All possible combinations | cayuse plaything, Indian pony plaything, horse plaything, Equus caballus plaything, pony plaything, cayuse water pistol, Indian pony water pistol, horse water pistol, Equus caballus water pistol, pony water pistol, cayuse hobby, Indian pony hobby, horse hobby, Equus caballus hobby, pony hobby, cayuse rocking horse, Indian pony rocking horse, horse rocking horse, Equus caballus rocking horse, pony rocking horse, cayuse slingshot, Indian pony slingshot, horse slingshot, Equus caballus slingshot, pony slingshot, cayuse catapult, Indian pony catapult, horse catapult, Equus caballus catapult, pony catapult, cayuse toy, Indian pony toy, horse toy, Equus caballus toy, pony toy |
| With SBERT cos sim > 0.8 | pony toy, pony plaything, Indian pony toy, horse toy |

a single caption named "pony toy". Not all combinations generated by substituting each noun in the phrase by its synonym is a valid combination or even meaningful phrase in the English language.

## A.5 USING SEMCLIP EMBEDDING FOR DEPLOYING IN CLOUD VECTOR STORES

We can use SemCLIP image-text embedding for deployment in any vector store as follows. We initialize the textual embeddings of the vector store with all nouns and text captions used during training to serve as initial vocabulary. Any new text caption acquired during subsequent deployment can be added as an average vector formed from its constituent nouns. An incoming image file $I$ is mapped to a vector $C_i' = \Gamma_t(C_{t_i})$ where $t_i = \arg\min_{t'} d(C_i, C_{t'})$ where $d$ is the cosine distance between the image and text vectors in the original CLIP space $C$ as explained in Section 3. A new query $Q$ is projected into SemCLIP directly through the semantic text embedding of its composed entities as $C_q'$. The nearest images to $Q$ are retrieved within the neighborhood of $C_q'$ using cosine similarity in the SemCLIP space.

## A.6 ADDITIONAL DETAILS ON THE RESULTS OF TABLE 5

In Table A.6 we provide more details on our image-to-text and text-to-image retrieval performance using additional measures of precision and recall for topK=5 results.

Table 11: Illustration of results of image and text retrieval using precision recall metrics.

| Dataset | Images / Labels | Model | Recall@10 (t2i) | Precision@10 (t2i) | Recall@10) (i2t) | Precision@10 (i2t) |
|---|---|---|---|---|---|---|
| **Visual Genome** | 7794 / 14513 | **SemCLIP** | **0.182** | **0.169** | **0.273** | **0.192** |
| | | CLIP | 0.065 | 0.022 | 0.027 | 0.045 |
| | | OpenCLIP | 0.080 | 0.027 | 0.035 | 0.055 |
| | | BLIP | 0.087 | 0.031 | 0.038 | 0.063 |
| | | NegCLIP | 0.077 | 0.027 | 0.034 | 0.055 |
| | | PosCLIP | 0.090 | 0.035 | 0.034 | 0.057 |
| **CUB** | 11788 / 200 | **SemCLIP** | **0.781** | **0.813** | **0.852** | **0.843** |
| | | CLIP | 0.084 | 0.497 | 0.826 | 0.083 |
| | | OpenCLIP | 0.110 | 0.656 | 0.935 | 0.093 |
| | | BLIP | 0.032 | 0.190 | 0.543 | 0.054 |
| | | NegCLIP | 0.065 | 0.386 | 0.694 | 0.069 |
| | | PosCLIP | 0.080 | 0.477 | 0.788 | 0.079 |
| **SUN** | 16657 / 567 | **SemCLIP** | **0.587** | **0.729** | **0.810** | **0.683** |
| | | CLIP | 0.191 | 0.384 | 0.595 | 0.059 |
| | | OpenCLIP | 0.259 | 0.522 | 0.633 | 0.063 |
| | | BLIP | 0.194 | 0.389 | 0.556 | 0.055 |
| | | NegCLIP | 0.201 | 0.404 | 0.567 | 0.056 |
| | | PosCLIP | 0.213 | 0.427 | 0.589 | 0.058 |
| **AWA2** | 6985 / 10 | **SemCLIP** | 0.971 | 0.980 | 0.989 | 0.912 |
| | | CLIP | 0.015 | **0.99** | 1.0 | 0.1 |
| | | OpenCLIP | 0.016 | **1.0** | 1.0 | 0.1 |
| | | BLIP | 0.016 | **1.0** | 1.0 | 0.1 |
| | | NegCLIP | 0.016 | **1.0** | 1.0 | 0.1 |
| | | PosCLIP | 0.016 | **1.0** | 1.0 | 0.1 |
| **COCO** | 5000 / 80 | **SemCLIP** | **0.921** | 0.89 | **0.889** | **0.823** |
| | | CLIP | 0.081 | 0.796 | 0.744 | 0.195 |
| | | OpenCLIP | 0.086 | 0.846 | 0.799 | 0.211 |
| | | BLIP | 0.089 | 0.877 | 0.781 | 0.204 |
| | | NegCLIP | 0.082 | 0.811 | 0.808 | 0.214 |
| | | PosCLIP | 0.087 | 0.91 | 0.843 | 0.230 |
| **Flickr30k** | 31014 / 158391 | **SemCLIP** | 0.581 | **0.572** | **0.567** | **0.584** |
| | | CLIP | 0.509 | 0.051 | 0.319 | 0.162 |
| | | OpenCLIP | 0.588 | 0.059 | 0.382 | 0.193 |
| | | BLIP | **0.725** | 0.072 | 0.480 | 0.243 |
| | | NegCLIP | 0.591 | 0.059 | 0.360 | 0.182 |
| | | PosCLIP | 0.468 | 0.047 | 0.283 | 0.143 |

