# OpenReview forum: "SemCLIP: Aligning vision-language encoder models to semantic spaces for stability in retrieval"
_ICLR.cc/2025/Conference — Submitted to ICLR 2025_

### Official Review · Reviewer_GR5C · 2024-10-30

**Soundness:** 3
**Presentation:** 3
**Contribution:** 3
**Rating:** 6
**Confidence:** 4

**Summary:**

This paper investigates the semantic matching of vision and language embedding vectors. Specifically, textual synonyms should match with similar images, but current vision-language models like CLIP often fail to retrieve similar images for synonymous terms. To address this issue, the paper proposes SemCLIP, which includes a new database of linguist-curated similarity lists of words. Additionally, it trains semantic-preserving textual embeddings (STE) to capture synonym relationships and aligns vision-language embeddings with STE to produce semantic-preserving embeddings for improved retrieval.

**Strengths:**

1. This paper addresses an important challenge in current vision-language embedding models, where synonyms with the same semantics are not well-mapped within vision or language embeddings.

2. Releasing the synonym-focused dataset as an open-source resource would be a valuable contribution to the community.

3. The evaluation on retrieval using images and labels (measured by mAP and NDCG) is also meaningful, providing a more nuanced comparison than traditional image-to-text matching, which may involve overlapping "same" semantic text.

**Weaknesses:**

1. Figure 2 is somewhat difficult to understand; it would be helpful to include explanations of each component (A, B, C, D, and J1, J2) in the caption.

2. It would be beneficial to report standard image-to-text or text-to-image retrieval metrics, such as Recall scores on MS COCO and Flickr, for comparison with existing CLIP methods.

3. The notations in the methods section are complex; it is unclear at times whether they refer to data samples or embedding vectors. Clarifying these distinctions would improve readability.

**Questions:**

I wonder if general CLIP models with larger backbones and training on large-scale datasets (in the billions), such as CLIP ViT-bigG, might not suffer from the synonym problem, as they are more likely to include diverse examples with synonyms in training and may be sufficiently generalized to handle synonym issues. Do you have any experimental results with these stronger VLM models?

---

> ### Author Response · Authors · 2024-11-22
> **Response to reviewer GR5C**
>
> 1. Figure 2 is somewhat difficult to understand; it would be helpful to include explanations of each component (A, B, C, D, and J1, J2) in the caption.
>
> >> The description of the Figure  2 was given in Section 3 (Lines 165-172, 245-247). However, we have added additional clarification in the caption of Figure 2.
>
> 2.  It would be beneficial to report standard image-to-text or text-to-image retrieval metrics, such as Recall scores on MS COCO and Flickr, for comparison with existing CLIP methods.
>
> >>The NDCG and MAP are more comprehensive metrics for search that subsume precision and recall numbers within. However, we have provided the precision and recall @10 similarly for the various datasets and methods in Table 11 in Appendix A.6 due to the space limitations in Table 5 of the paper. If requested we can try to squeeze those into Table 5. Based on the precision and recall numbers as well, SemCLIP can be seen to outperform SOTA as well.
>
>
> 3. The notations in the methods section are complex; it is unclear at times whether they refer to data samples or embedding vectors. Clarifying these distinctions would improve readability.
>
> >> We have improved the notation in Section 3 of the revised draft and making them more consistent to distinguish between data samples and vectors. Please see revised version.
>
> 4. I wonder if general CLIP models with larger backbones and training on large-scale datasets (in the billions), such as CLIP ViT-bigG, might not suffer from the synonym problem, as they are more likely to include diverse examples with synonyms in training and may be sufficiently generalized to handle synonym issues. Do you have any experimental results with these stronger VLM models?
>
> >> The basic premise of our approach was that text embeddings used to build VLMs have not incorporated synonyms directly but infer them primarily from use context. So we expect such limitations to be there for any models build with text transformers underneath. To illustrate this, we ran our experiments of Table 4 testing overlap ratios of synonyms using the CLIP VIt-BigG model (hf-hub:laion/CLIP-ViT-bigG-14-laion2B-39B-b160k) and still saw the same phenomenon. This result is now added to Table 4 under CLIP-G. Although the overall performance of CLIP-G is better than the models tested, it has a huge memory footprint. Yet our SemCLIP outperforms these as can be seen from Table 4.
>
> >> Figure 1 is now replacing CLIP with  CLIP VIt-BigG  results to illustrate the semantic recognition of synonyms is still absent in the bigger models as well when the synonyms of the word 'basket' bring different top results for each synonym (Figure 1 a-d). However, the average vector approximation of nouns holds similarly for bigger models as well as seen from Figure 1 e-g.

---

> > ### Comment · Reviewer_GR5C · 2024-12-01
> >
> > Thanks for the rebuttal which helps me better understanding of the paper.

---

> > > ### Author Response · Authors · 2024-12-02
> > > **response**
> > >
> > > We are glad you found the rebuttal helpful. Let us know what else you would like us to furnish to result in improved scores.

---

### Official Review · Reviewer_wCVL · 2024-11-02

**Soundness:** 2
**Presentation:** 2
**Contribution:** 2
**Rating:** 5
**Confidence:** 4

**Summary:**

The paper focuses on establishing stable association between images and texts, to make synonymous queries bring up the same images or have a high degree of overlap, and propose a SemCLIP framework, which consists of two main step, semantic text embedding and alignment transform. Besides, the paper develops a database of linguists-curated similarity lists of words. Performance comparison on multiple benchmark datasets show the effectiveness of the proposed framework.

**Strengths:**

The paper proposes a framework to transform the VLM embeddings of semantically close terms and their associated images to place close together to ensure retrieval stability, which is practical for vector-database-based data managing.

**Weaknesses:**

1. The novelty is limited. The proposed SemCLIP framework is mainly composed of semantic text embedding and alignment transform. However, they are all implemented through existing simple algorithmic ideas, without introducing in-depth perspectives on semantic alignment.

2. In Modeling SemCLIP transformations (page 4), in the computation of alignment transform of images (i.e., equation 4), the determination of image representations replies on the distance between the image and its nearest text in VLM space. This implies that SemCLIP acknowledges the validity of relative distance between images and texts in VLM space, which conflicts the main claim about the loss of sensitivity to linguistic similarity in VLM space, in paper.

3. The writing should be improved. First, the notation in this paper needs further refinement and standardization. Secondly, some of the statements in paper are confusing and need more in-depth explanation. For example, lines 192-193 “Note that the distance is a function of the word itself, since some words have more synonyms than others.”, lines 290-291 “We developed a contrastive embedding model that captures the essence of the similarity in the curated similarity lists in a numerical formulation”, and so on. Thirdly, The implementation of semantic text embedding and alignment transform in section 4, 5 and the previous section 3 seem to be conceptually separate, and need further re-organization.

**Questions:**

See Weaknesses.

---

> ### Author Response · Authors · 2024-11-22
> **Response to Reviewer wCVL**
>
> 1. The novelty is limited. The proposed SemCLIP framework is mainly composed of semantic text embedding and alignment transform. However, they are all implemented through existing simple algorithmic ideas, without introducing in-depth perspectives on semantic alignment.
>
> >>
> Our paper throughout is giving several in-depth perspectives on semantic alignment. It is showing that the lack of semantic consistency seen in retrieval is due to a problem with textual embeddings which are trained in a data-driven manner and hence unable to infer the meaning similarity between two words that don’t occur in the same sentence to utilize usage context. We are also avoiding training or fine-tuning VLM but able to considerably increase performance through the use of a semantic alignment transform computed a priori. Our novel contributions are in (a) similarity list dataset for all nouns (b) STE embedding with main downstream uses (c) alignment transform to make VLMs semantically consistent and stable w.r.t synonyms
>
> 2.  In Modeling SemCLIP transformations...This implies that SemCLIP acknowledges the validity of relative distance between images and texts in VLM space, which conflicts the main claim about the loss of sensitivity to linguistic similarity in VLM space, in paper.
>
> >>The loss of sensitivity we are referring to is the inherent inability of textual embedding of synonymous words to be close together. We believe that CLIP’s joint embedding does bring an image and textual captions close together by design. However, this is done pairwise for the respective textual captions. In other words, similar textual captions are not sharing similar nearby images. This is what we showed in Figure 1 through an example, and Table 2 showed this through experimental results. Hence these are not conflicting with our claim.
>
> 3. The writing should be improved.
>
> >> The paper has been revised and section re-arrangements done for better reorganization. Please see the revised version.

---

> ### Comment · Reviewer_wCVL · 2024-11-25
>
> I still maintain that this paper lacks in-depth research perspective and keep my score 5 unchanged.

---

> > ### Author Response · Authors · 2024-11-25
> > **Clarification sought**
> >
> > Could you please elaborate through an example what additional details you are looking for  "in-depth research perspective". This would be helpful for us to respond with further information. Note that for many of the terms mentioned in your comment, we have revised the description in the paper as well as given more information in Appendix on the complexity of semantic modeling of language elements through numerical formulations where the ontological relationships are being rendered through neural embeddings.

---

### Official Review · Reviewer_P8Vf · 2024-11-03

**Soundness:** 2
**Presentation:** 2
**Contribution:** 3
**Rating:** 5
**Confidence:** 4

**Summary:**

This paper addresses the problem of image retrieval with synonymous text. It develops a dataset of linguists-curated similarity lists of
words and trains a semantics-preserving textual embedding (STE) to which the VLM embedding is aligned. Experiments on 13 benchmark datasets demonstrated the effectiveness of the proposed method.

**Strengths:**

1. This paper is well presented, with clear figures and organizations.
2. This paper develops a dataset for synonymous text understanding.

**Weaknesses:**

1. The description of how to construct the similarity list dataset is not clear enough.
 - It would be better to give out a figure for a more intuitive illustration.
 - Is the synonymousness defined for both nouns and phrases? Is the similarity defined with binary values (0/1) or continuous values between 0-1? Any difference?
 - It is concerned that are there any cases in two worlds with the same meaning on single nouns but different meanings in sentences?
 - How to ensure that the items in the dataset are diverse enough, reasonable, and frequently used?
2. Lack of important experiments.
 - Does the learning of synonymous text degenerate the retrieval of general words? Tables 2 and 3 only show the results of querying using synonyms of nouns but no general words.

**Questions:**

1. From methodologies, it seems that only the text encoder is updated for better alignment between synonymous pairs, while in Figure 2, image embeddings of J1 and J2 are pulled, how? Is the original VLM also fine-tuned?

---

> ### Author Response · Authors · 2024-11-22
> **Response to Reviewer P8Vf**
>
> 1. The description of how to construct the similarity list dataset is not clear enough.
> It would be better to give out a figure for a more intuitive illustration.
> >> We have added further description on this in Appendix and Figure 4 to explain the similarity list construction process in Section A 2.2. Illustration of similarity lists was already provided in Table 6 in Section A 2.2. Note that the items in a similarity lists are not the pure synonyms but also hypernyms based on the starting lists in the ontology obtained by distance-based navigation in the Wordnet ontology using the WUP similarity metric
>
> Is the synonymousness defined for both nouns and phrases? Is the similarity defined with binary values (0/1) or continuous values between 0-1? Any difference?
>
> >>Our collection is based on what Wordnet classified under the noun class. Many of them include phrases and noun phrases, e.g. “ejector seat”, “decaffeinated coffee”, “fire hydrant”, “old man of the mountain”,  etc.
>
> >> The similarity initially was computed using the Wu-Palmer metric which is a numeric range from 0-1 as referred to in Equation 6 in the paper. However, once the curated similarity lists are formed, then all synonyms of an anchor word are treated equally in the contrastive learning formulation where they are all taken as positive examples. The revised architecture diagram of Figure 3 also shows this more clearly.
>
> It is concerned that are there any cases in two worlds with the same meaning on single nouns but different meanings in sentences?
>
> >> Yes, it is possible to have multi-sense words. These are words that acquire meaning in a context. Wordnet has already catalogued all these senses in its synset notation. Our STE embedding explicitly accounts for all these senses so that a single core noun in plain English form has different lemma forms and it is these lemma forms that are encoded. Please see example in  A.2.1
>
> How to ensure that the items in the dataset are diverse enough, reasonable, and frequently used?
> >> By capturing the entire English language nouns in our vocabulary as per Wordnet, we have tried to ensure that the similarity list dataset (which will be contributed to open source) is sufficiently diverse. In addition, the alignment transform used nearly 800,000 captions that are also commonly occurring.
>
> 2. Does the learning of synonymous text degenerate the retrieval of general words? Tables 2 and 3 only show the results of querying using synonyms of nouns but no general words.
>
> >> The results in Table 5 are covering ‘general words’ (termed labels in column 2 of Table 5). This shows that not only does our method cover synonyms well, but because it has a better understanding of the meaning of words, can perform general downstream tasks of text-to-image and image-to-text retrieval as well zero-shot classification, all of which are tested with ‘general words’ from various collections. Table 5 alone shows the results with (14513(visual genome)+200(CUB)+567 (SUN)+10(AWA2)+80 (Coco)+158391(Flickr30k)) words that may or may not have their synonyms in the label set.
>
> 3. From methodologies, it seems that only the text encoder is updated for better alignment between synonymous pairs, while in Figure 2, image embeddings of J1 and J2 are pulled, how? Is the original VLM also fine-tuned?
>
> >> Equation 4 indicates the alignment transformation for images J1 and J2.
>
> >> The key idea exploited here is that since CLIP is already trained to bring the images closer to text embeddings, the alignment done using text embedding can be carried to their corresponding nearby images. Equation 3 shows the projection operation for both images and text. It is the alignment that is designed from scratch as is the STE embedding. In a sense, the alignment is a way of fine-tuning the VLM by ‘fixing’ it.

---

### Official Review · Reviewer_hb5v · 2024-11-04

**Soundness:** 3
**Presentation:** 3
**Contribution:** 3
**Rating:** 6
**Confidence:** 4

**Summary:**

The authors developed A database of 114,000 linguists-curated similarity lists of words from a constrained traversal of Wordnet thesaurus to cover all English language nouns and use a representation to capture their sense context explicitly. And then a semantics-preserving textual embedding was trained to discover expanded synonymous relations between terms.  A method was developed to align a VLM embedding.

**Strengths:**

1. This paper identify and address the issue of instability in VLMs when dealing with synonymous queries.

2. The authors develop a dataset of linguist-curated similarity lists , followed by an alignment transformation to map existing VLM embeddings to the semantics-preserving textual embedding.

3. Abound experiments provides extensive empirical evidence to support the effectiveness of SemCLIP model, including comparisons with multiple benchmark datasets and other CLIP variants.

**Weaknesses:**

1. The SemCLIP model is developed only for nouns in the English language. This limitation narrows the applicability of the model to other parts of speech.

2. The database of similarity lists, while valuable, may introduce bias based on the linguists' perspectives and may not capture the diversity of language use across different domains.

3.  WSD tool has an accuracy rate of around 80%, which could introduce errors in the alignment process.

**Questions:**

1. The alignment transform of image embeddings from VLM to SEM space is confusing. Image embedding and text embedding are directly equated, which may overlook the semantic differences between images and text.

2. How to determine the distance of semantic similarity, gamma_j ?

3. The Transformation Mapping stage in Fig. 3 lacks arrows connecting to other stages, making it difficult for the reader to intuitively understand how the output vector is utilized in the subsequent stages.

4. Why STE perform worse in antonyms dataset?

5. Long-tail or rare synonyms that may not be well-represented in WordNet could affect semantic text embedding.

---

> ### Author Response · Authors · 2024-11-22
> **Responses to comments**
>
> 1.The SemCLIP model is developed only for nouns in the English language. This limitation narrows the applicability of the model to other parts of speech.
>
> >>Approximating a full query sentence by only the nouns within it gives similar results to that of using the full query. This was illustrated qualitatively in Figure. 1 (e,f,g) in which the full sentence was replaced by only the nouns within. Quantitatively also we have shown in Section A.1 and Figure 4 in appendix. The result there indicates that the approximation of full natural language query phrase by nouns alone gives nearly 90% equivalence in the top matches (the original caption was the nearest vector for 90% of the average vectors with an average cosine similarity of 0.958). Both these results indicate that restricting to nouns is not a severe limitation for our application as most people search for objects in images.
>
> >>The restriction to nouns was also done to use a manually validated list of similarities in building our STE embedding. Such arduous curation is underway for other parts of speech by our linguists and will be similarly made available in future.
>
> 2.	The database of similarity lists, while valuable, may introduce bias based on the linguists' perspectives and may not capture the diversity of language use across different domains.
>
> >>
> 1.	We observe first that Wordnet is fairly comprehensive in coverage of the diversity of language used to refer to terms including slangs, old English words, and several domain-specific terms. In comparison to word vocabulary used for constructing language models such as BERT variants (30,000-50,000 words), we are already using all nouns (over 140,000 words) vocabulary which should only add to the diversity.
> 2.	Queries used for our experimental results in Table 5 and 6 came from a vocabulary derived from publicly available datasets which was intended to cover the diversity of language across domains as well. For example, the CUB dataset is specialized on bird species vocabulary.
> 3.	However, to avoid linguistic bias that cause ordinary meanings of words  to be chosen over domain-specialized meanings, we propose that STE embeddings be similarly developed for such vocabularies starting from the ontologies in those domains and be used to prioritize these vocabularies during search. Our method clearly specifies how such embeddings can be developed by a 3 step process of (a) traversing the ontological relationships of synonyms and hypernyms, (b) Forming initial similarity lists based on distance between nodes, (c) curating the lists to retain essential items similar in meaning to anchor words. This workflow is also shown in Figure 5 in Appendix.
>
> 4.	WSD tool has an accuracy rate of around 80%, which could introduce errors in the alignment process.
> >>
> During training of the alignment model, the mapping was derived from the definition of nouns which avoid the WSD problem. Further, during query time, the averaging of the noun vectors mitigates some of the WSD errors. However, WSD research needs to be further pursued (see Appendix A3.2 for extended discussion on this in the paper). Finally,  even with the current accuracy of WSD, our method outperforms all SOTA methods.
>
> 5. 	The alignment transform of image embeddings from VLM to SEM space is confusing. Image embedding and text embedding are directly equated, which may overlook the semantic differences between images and text.
> >> The revised architecture diagram in Figure 3 of the revised submission shows the transformation process more clearly. The alignment transform starts from the joint embedding space vectors of CLIP which already accounts for the semantic similarity between images and text. In the joint embedding space, both image and text are 512-length vectors and indistinguishable from numerical standpoint. However, our projection process maintains the identity of the image and text vectors by first mapping images to text using CLIP and then applying the alignment transform.
> 6. Choice of threshold Gamma_j
> >>
> A value of gamma_j of 0.6  was chosen for Wordnet nouns  as the learned embedding exhibited sharp fall off for non-synonym neighbors after this value.
>
> 7. The Transformation Mapping stage in Fig. 3 lacks arrows connecting to other stages....
> >>
> The architecture diagram of Figure 3 is updated showing clearly the connections between the various components of SemCLIP model.
>
> 8. Why STE perform worse in antonyms dataset?
> >>
> This behavior is as expected since antonyms are opposites of the class being captured (synonyms) and hence should not match the anchor word.
>
> 9..	Long-tail or rare synonyms that may not be well-represented...
>
> >>  Wordnet already captures many obscure synonyms. While the average vector approach handles any out-of-vocabulary words including rare synonyms,  the STE embedding can be further fine-tuned  for such long-tailed synonyms whenever available.

---

### Meta-Review · Area_Chair_ndnU · 2024-12-20

**Metareview:**

This paper studies the semantic similar query embedding in the CLIP embedding space. The paper argues that the CLIP doesn't preserve the semantic similar text embedding. To solve this task, the paper proposed to collect a dataset contains synonyms from WordNet and adapt the CLIP embedding to aligned with the embedding space learnt from the WordNet synonyms dataset.

Strength:
1. the task of studies semantic similar query embedding in the CLIP embedding space (VLM embedding space) is important.
2. The paper collects a dataset which might be helpful for the future research.

Weakness:
1. Paper is quite hard to read. This is mainly due to the notations. The figure 2 didn't show why the suboptimal solution is suboptimal. From the figure, it is hard to understand why the proposal approach is better than the suboptimal one.
2. I shared the same concerns with reviewers on whether the synonyms from the WordNet is enough to cover the semantic similar query. In English, we are able to construct sentences which contain similar linguistic structure with words / nouns having similar meanings. Those sentences might not share the same semantic meaning or might even have totally opposite meaning. I also doubt whether the WordNet could capture the subtle differences in the semantic meaning.
3. The author only shows the Top-10 retrieval precision-recall score in page 18. This might not enough to guarantee the perfomrance on the general task. I would suggest the author to include additional metrics such as Top-1 accuracy and zero-shot imagenet classifcation score.

This is a borderline paper.  The reviewers didn't reach agreement on the final decision. As AC, based on my reading and understanding, I think the cons overweight the pros (especially the presentation is verbose and not very easy to understand.) Given the concerns and the weaknesses, I think this paper is not ready for ICLR in both presentation and the contribution. I would recommend reject.

**Additional Comments On Reviewer Discussion:**

During the rebuttal, all reviewers maintain their score. Before the rebuttal, the reviewers listed the concerns with presentation and the coverage of WordNet over the semantic similar queries. Neither the reviewer nor me were persuaded that those concerns were addressed. In the rebuttal, the author reiterate the points that the WordNet is sufficient without providing new evidence to support this claim.

---

### Decision · Program_Chairs · 2025-01-22

Reject